# Employed mothers' breastfeeding: Exploring breastfeeding experience of employed mothers in different work environments in Ethiopia

**Firmaye Bogale Wolde**[1]*, **Jemal Haidar Ali**[2], **Yalemwork Getnet Mengistu**[2]

**1** Department of Knowledge Translation, Ethiopian Public Health Institute, Addis Ababa, Ethiopia, **2** School of Public Health, Addis Ababa University, Addis Ababa, Ethiopia

* fbfbogale93@gmail.com

**Data Availability Statement:** The data that support the findings of this study are available from the corresponding author, (FB), and have also been provided as Supporting information files.

## Abstract

### Background

One of the most cost-effective interventions to enhance child health with the potential to reach families of all economic backgrounds is breastfeeding. Despite the many benefits optimal breastfeeding has, its practice is low due to various barriers among which maternal employment is mentioned repeatedly. Accordingly, this study has explored the experience of employed mothers with regards to breastfeeding, employment, and work environment in Addis Ababa Ethiopia.

### Methods and findings

A descriptive Phenomenology strategy was employed among 17 mothers drawn from different organizations, conveniently, that offer accommodation of six months maternity leave or onsite child care center or had only three months maternity leave. Data were collected through in-depth interviews until information saturation was reached. Recorded interviews were transcribed and translated and the information obtained was then organized and coded to generate overarching themes. Two themes on facilitators and barriers, and addressing barriers were generated after analysis. Mothers recognize the importance of breastfeeding for children but returning to work at three months is expressed as a major barrier to continuous breastfeeding. Mothers who have access to supporting conditions at their workplace expressed better breastfeeding practice and better satisfaction with their job.

### Conclusions

Providing employed mothers with a supporting environment helps them work with better stability, motivation, and satisfaction. This however requires a suitable accommodation with a focus on the different kinds of work environments of the mothers and the different risks related to each respective environment via scaling up and monitoring breastfeeding interventions and calling upon institutions to remove structural and societal barriers to breastfeeding.

**Funding:** The authors received no specific funding for this work.

**Competing interests:** The authors have declared that no competing interests exist.

## Introduction

Breastfeeding is a life-saving and one of the most cost-effective interventions to enhance child health with the utmost potential to reach families of all economic backgrounds [1,2]. It is highly recommended for a reduced risk of gastrointestinal infection, pneumonia, and otitis media which are some of the common causes of child morbidity and mortality in developing countries [3,4]. According to the World Health Organization (WHO) and United Nations International Children's Emergency Fund (UNICEF) recommendations, optimal breastfeeding indicates exclusive breastfeeding of a child for the first six months of life continuing to the age of two and beyond with appropriate and sufficient complementary foods [5,6].

The 2020 WHO report of infant and young child feeding reported that over 820,000 under five years' children could be saved every year if all children 0–23 months were optimally breastfed [1,2,5,7].

Furthermore, according to studies done on the association of breastfeeding and child mortality, neonatal, infant, and under-five child mortality are reported to be lower among breastfed children than non-breastfed children [8–11].

It is indicated that women's empowerment is an important element for a better breastfeeding practice, and one of the significant contributors to achieving this globally is women's employment [10,12–14]. But despite this, studies globally and in Ethiopia show that it contrarily contributes to poor breastfeeding practice where the prevalence of breastfeeding among employed mothers is lower than that of unemployed mothers due to various work-related factors [15–18]. Some of the factors that could serve as both barriers and facilitators according to the way of their application in the workplace include social support in the workplace, support from workplace supervisors, length of maternity leave, amount and distribution of work time, flexible scheduling at work, on-site child care, type of equipment and physical layout of the workplace like breast pumps and pumping stations and breastfeeding policies [19,20].

Studies in Ethiopia have also supported that factors like short duration of maternity leave, lack of flexible work time, and lack of lactation break are barriers to breastfeeding while employed. It is also shown that the support provided for breastfeeding women in their work environment is critical in supporting their breastfeeding practice [21–23].

Considering these, the Lancet breastfeeding series states that different actions are taken to increase the rate of exclusive breastfeeding (EBF) for the first six months to 50% in 2025 from 37% in the year 2016 [7,24]. These actions include scaling up and monitoring breastfeeding interventions and calling upon institutions to remove structural and societal barriers to breastfeeding. Accordingly, interventions including longer maternity leave, onsite childcare center (CCC), breast pump, and lactation room are commonly suggested ways of creating mother-friendly worksites [25,26]. Even though there are some efforts to make workplaces mother-friendly by constructing onsite child care centers the effort is still minimal in Ethiopia. Additionally, Ethiopia has increased maternity leave from 90 days to 120 days incorporating 30 days prenatal and 90 days postnatal leave in September 2019 [27] which is still below the 18 weeks maternity leave recommendation of the international labor organization [28].

There are different quantitative studies done on breastfeeding and employment showing factors and their associations [18,21,29]. However, few focus on exploring the opinions and lived experience of employed mothers with regards to breastfeeding, employment, and supporting conditions in the workplace qualitatively [30]. Consequently, the objective of this study was to understand the motivations, barriers, and coping mechanisms to breastfeeding in employed mothers in the presence or absence of supporting conditions at their workplace.

Thus, this study explored the breastfeeding experience of employed mothers in Addis Ababa, Ethiopia which showed the experience as well as the standing point to further investigate the issue and find an appropriate solution.

## Materials and methods

### Ethical consideration

The study received ethical approval from the Research Ethical Committee of the School of Public Health, Addis Ababa University. Following this, written informed consent was obtained from the participants after the explanation of the purpose, procedures, benefits, and risks of the study were made. All the interviews were done with strict privacy.

### Research questions and objectives

This study had two main research questions which are: (1) what does the experience of breastfeeding for employed mothers look like? (2) How does the breastfeeding experience differ between employed mothers who have six months of maternity leave or access to an onsite childcare center or have neither of the two? Accordingly, the specific objectives of this study were exploring the breastfeeding experiences of employed mothers in different working environments with different supporting arrangements, and identifying and understanding motivations, barriers, and mechanisms of addressing challenges to breastfeed while employed in different working environments.

### Design

A descriptive Phenomenology strategy was employed in this study where understanding the lived experience is marked as a philosophy as well as a method. This procedure involves studying a small number of subjects through extensive engagement to develop patterns and relationships of meaning as described in Creswell's research design 3rd edition [31].

This strategy is used in this study to address the objectives of the study by helping explore and understand the lived experience of mothers related to breastfeeding and employment. Furthermore, the application of this design helps to apprehend the subjective experience of employed mothers and to gain insight into their perception, emotion, thought, and action giving chance to development of better recommendations.

### Participants

The study population in this study included full-time women employees working in any position and have a currently breastfeeding child of two years or younger. These women were employed in an institution that offers a six months maternity leave, or in an institution that has an onsite childcare center, or in an institution that has neither. All participants were from urban areas and lived in a highly crowded city.

A purposive selection was used to choose institutions in Addis Ababa city taking their provision of the different supporting environment, operationally defined in this study as institutions that have six months of maternity leave and an onsite childcare center in the work environment, for breastfeeding employees and those with only three months maternity leave into consideration. The representation of private, governmental, and non-governmental organizations was also considered in the selection.

Eligible participants were identified purposively using predefined inclusion criteria where responsible personals, mostly HR officers, in each organization helped to identify participants

in each institution. After identification, eligible participants were contacted by researchers if only they have expressed their willingness to be approached.

## Data collection

Data were collected between Dec 2016-May 2017 in Addis Ababa, the capital city of Ethiopia, in governmental, private, and non-governmental organizations. The included organizations had three months of maternity leave or six months of maternity leave or an onsite childcare center. Data were from women who met the inclusion criteria and were collected until information saturation was attained which was determined by the repetition of information and lack of newly emerging information.

Data collection was carried out by adapting a national breastfeeding study in Pakistan which has a section of interviews for employed breastfeeding mothers and is used in countries like Uganda, Malawi, and Senegal [32]. This protocol was edited to fit into this study considering the objective and the study population.

The interview guide contains semi-structured and a few open-ended questions to have a high degree of flexibility. The demographic data on the age of the mother, marital status, working position, Years of schooling, the total number of children, and the gender of the youngest child, and the age of the child were collected. Additionally, each interview guide had core questions that did not direct the participants and permitted them to express what they felt was important while maintaining a central frame of inquiry. The interview also included questions regarding general knowledge on breastfeeding, breastfeeding experience of mothers concerning employment, and experience of breastfeeding with regards to supporting conditions in the workplace. Some are:

- For how long do you think a child should be breastfed?

- Can you please tell me what it was like for you to breastfeed after returning to work?

All interviews were conducted by the principal investigator (FB) who is a public health professional by training and worked in a governmental organization where she has worked with employed mothers but did not have experience of her own on breastfeeding and was not particularly very knowledgeable on the issue.

The date, time, and place of each interview were arranged following what was most favorable and comfortable for the subjects. Accordingly, interviews were conducted in the mothers' offices. All the interviews were audiotaped with the participant's consent and the interviews took from 40 to 60 minutes. Informed consent was taken before each interview where participants were informed about the purpose, procedures, benefits, risks, and the right of the respondents to refuse to answer a few or all of the questions.

## Data analysis

Thematic data analysis was performed to describe and compare general statements as relationships and themes present on the data according to the method of analysis described by Kleiman (2004).

All three authors were involved in the preparation of this study where FB was involved starting from the conception of the idea of the study and original article write-up to collecting data and analysis to final write-up. JH has been involved as a senior researcher starting from guiding the study conduction and supervision to involving in the analysis of the study from coding to writing up of the study and the third author YG has been involved in conception and analysis from coding to writing up and editing of the study. All interview transcripts were

read and counter-checked with the transcripts, and emerging themes were agreed upon after coding.

The interviews of this study were digitally recorded, transcribed verbatim, and translated to English from the local dialect of Amharic.

The first stage of data analysis involved closely reading and re-reading each transcript and listening to the audio recording to become familiar with the data.

The second stage involved initial coding by reading the transcript in-depth and making a note of anything important or interesting and paying attention to where and how patterns occurred.

The third stage was categorizing codes into themes directed by the content of the data.

In the fourth stage searching for connections across emergent themes and checking if the themes make sense and account for all the coded extracts and the entire data set was done.

The fifth stage involved generating clear definitions and names for each theme and describing which aspects of data are being captured in them and that they had meaningful contributions to answering the research question by systematically going through the transcripts of each one of the 17 participants.

During data collection member checking was done by restating and paraphrasing information for respondents to determine the accuracy and participants were revisited one more time for clarity of information and accuracy of meaning before data analysis. Reliability was ensured by trying to avoid leading questions and all Amharic transcripts were cross-checked with the oral discourse for consistency and completeness of data and interview transcripts and codes were constantly compared with the data to prevent drift of definition of codes (Gibbs 2017). The analysis was done manually and rigor was enhanced through regular discussions between all three authors.

## Results

### Mothers' characteristics and working environment

The general characteristic of participants in this study is presented in (Table 1). Out of 17 mothers in this study 11 had only one child whereas five mothers had two children and only one mother had four children. The mean age of the mothers was 30 and the mean age of the index child was 13months old. Participants were full-time employed mothers from five organizations where seven were from governmental, five from private, and the rest five from non-governmental organizations. Most of the included mothers (seven mothers) had a first degree followed by six mothers who had a diploma where the highest educational level was a master's degree held by one mother.

Out of the organizations included in this study, one provided six-month maternity leave and two organizations had onsite CCC, and the rest two provided three-month maternity leave only.

The organizations that provide onsite CCC had their own back and forth transportation service for mothers. They also gave the mothers breastfeeding breaks for two sessions, lasting for 30 minutes each in addition to their lunch break.

There is a setting difference between the two organizations that have onsite CCC. One of the organizations was governmental and children in the center were cared for by babysitters and they could stay up to the age of five years. The second one was a private organization that trained mothers of the children and arranged for the mothers themselves to care for their children with a specific schedule one day per week. Thus, the organization didn't hire extra manpower to babysit and children could stay in the center up to the age of four years.

**Table 1. Employed mothers' characteristics in the study.**

| Employed Mothers characteristics | | | | | |
|---|---|---|---|---|---|
| Code | Age | Educational status | Occupation | Index child age in months | Organization |
| Mn | 28 | High school graduate | Janitor | 15 | Governmental |
| Mn | 30 | Diploma | Documentation officer | 11 | Governmental |
| Mn | 28 | Diploma | Finance officer | 12 | Governmental |
| Mn | 26 | First degree | Finance officer | 22 | Private |
| Mn | 31 | First degree | HR officer | 15 | Private |
| Mn | 25 | First degree | Health officer | 11 | Governmental |
| Mn | 26 | First degree | Pharmacist | 18 | Governmental |
| Mn | 27 | First degree | Professional nurse | 15 | Governmental |
| Mcc | 27 | Diploma | Machine operator | 12 | State enterprise |
| Mcc | 28 | Diploma | Secretary | 4 | Private |
| Mcc | 30 | Certificate | Guard | 7 | Private |
| Mcc | 32 | Diploma | Cashier | 10 | Private |
| Msm | 29 | First degree | Finance assistant officer | 14 | NGO |
| Msm | 34 | Diploma | Receptionist | 16 | NGO |
| Msm | 37 | First degree | Accountant | 22 | NGO |
| Msm | 38 | Master's degree | Monitoring officer | 8 | NGO |
| Msm | 31 | Diploma | Cashier | 9 | NGO |

Mother with no supporting environment (Mn)–an employed mother who had three months maternity leave but no onsite child care center nor six months maternity leave.

Mother with child-care center (Mcc)–an employed mother who had access to childcare center at their workplace.

Mother with six months maternity leave (Msm)–an employed mother who had six months maternity leave.

## Themes

After analyzing the collected data two themes emerged

1. Workplace barriers and facilitators to breastfeeding,

2. Addressing barriers

## Theme one- workplace barriers and facilitators to breastfeeding

Mothers mentioned diverse facilitators and barriers to breastfeeding. They explained their physical and emotional challenges to continue breastfeeding after returning to work by relating their experience with and view on their working environment.

**Subtheme one- past experience.** One of the factors majorly expressed as a motivator is a mother's experience of breastfeeding. Mothers talked about the difference in the level of motivation and satisfaction they get while breastfeeding when compared to their previous experience of not breastfeeding. They also explained the difference in the health status of their breastfed and non-breastfed child.

"*The fact that I didn't breastfeed my older baby pushed me to breastfeed my younger one. My older baby used to get sick all the time and I used to take her to the clinic repeatedly. But thanks to God my younger baby is healthy and I don't take her to the clinic often.*"

*Mn*

Participants also expressed a feeling of achievement and happiness as a key motivator to breastfeed

"*As I've told you before it (breastfeeding) is good for my child's health and growth and I'm happy because I feel like I've fulfilled my responsibility as a mother.*"

*Mn*

**Subtheme two- work challenges.** Mothers talked about several challenges they face when trying to continue breastfeeding while working.

Mothers experience emotional stress, worry, and instability in their daily endeavors. They explained that they have no choice except to accept the reality which sometimes pushes them to the point of giving up on their job.

"*I can't put my whole focus on work; I get frustrated because my child doesn't drink well and I think he might get hungry or cry while I am here. I feel sad and sometimes even want to quit my job because I've no choices.*"

***Mn***

Mothers with either of the supporting conditions at work i.e. six months of maternity leave or onsite CCC explained the fact that the tension of starting work is easier because of the supporting conditions they have at work.

"*Before I brought my child to the center, I was stressed about how she would stay the whole day without breastfeeding and I was worried that she might fall or get hurt. Everything about her made me stressed but it all stopped once I brought her to the center and everything got better.*"

***Mcc***

Mothers expressed the physical challenges they face when they are on their jobs and could not breastfeed. They explained breastfeeding after work is physically demanding and causes discomfort which harms their job.

"*After I spend my day at work I get tired and breastfeeding after that seems like another tiring job to do which even makes me mad. The other thing is after I spend the whole day at work my breast becomes full and becomes painful to even breastfeed while the breast is engorged.*"

***Mn***

**Subtheme three- experience difference.** The lack of supporting conditions at the workplace and its effect is strongly mentioned by mothers who have neither CCC nor six months of maternity leave. They have indicated that expressing milk to feed a baby at home would have been easier had there been a refrigerator in the office.

"*I spill my breast milk in the toilet when it gets full. It will be contaminated even if I express it here because there is nothing to store my milk in.*"

***Mn***

Mothers with neither six months of maternity leave nor an onsite childcare center expressed that having a supportive boss helped them in their breastfeeding habit and continuation of work.

"*I, fortunately, had a good boss and he used to let me go home and breastfeed with some intervals and I knew he would allow me to do that. If that was not the case I would quit my job.*"

*Mn*

Mothers who have access to onsite CCC expressed the presence of onsite CCC and additional breastfeeding breaks as a great support to breastfeeding.

"*We have breastfeeding breaks to breastfeed and check on our children. We come here at 10 am and 3 pm to breastfeed our children and we get back to work after 30 minutes. The presence of such a situation has made the problems easier.*"

*Mcc*

Mothers with six months of maternity leave expressed that the length of the maternity leave has helped them a lot and they are given their annual leave, in addition, which helped them with the continuation of breastfeeding.

"*There is no problem with the leave here. I even took additional leave after I finished my maternity leave and I stayed home for around 7 months. We are even allowed to come in late at 2 in the afternoon, 30 minutes more for breastfeeding.*"

*Msm*

A feeling of being treated differently after getting back to work is explained by mothers as everyone wants to extend their stay at home and this was done with the degree of closeness with their bosses.

"*After I got back to work I used to go out for half-day to breastfeed which was, of course, unofficial and done with only agreeing with my boss as most mothers do. But the difference is that mine lasted shorter. And later I was told that it will be deducted from my annual leave which is not the case for some other mother but I don't understand why such differential treatment prevails.*"

*Mn*

**Subtheme four- supporting condition at work.** Mothers who have neither of the two supporting conditions at work expressed the effect of the three months maternity leave on their experience of breastfeeding.

They explained that they had to start formula milk early because they had to get back to work early.

"*I stayed home for three months and breastfed my baby well. But after I resumed my work I started giving her formula milk which I don't think is healthy. But I had to since I didn't have additional support.*"

*Mn*

One of the major ideas explained by mothers who have either of the supporting condition was the difference of experience they have with their current baby and the older ones they had before working in an organization that offers these supporting conditions.

"*I previously left my children home at three or four months with a maid and I used to worry about them a lot. I had to give my babies formula milk early to get them used to it because I worry they might get hurt if I don't. But it is different now.*"

*Msm*

Mothers who have access to onsite CCC or who have six months' maternity leave talked about how lucky they are since they got the option to continue breastfeeding.

"*The care provided in this organization is immense. What this organization offered has enabled us to provide more care for our kids since we have the opportunity to see our children during our working hours. This shows how much we are valued by our employer.*"

*Mcc*

Mothers from organizations that have onsite CCC expressed a slightly different opinion on the way children are cared for. Those in the institution that hires babysitters to take care of children said their concern is that the babysitters sometimes become busy and tend to compromise childcare.

"*I'm happy about this center but the one concern I have is that the nanny here is busy because she takes care of all the children and becomes tired. I think she needs help and we have asked for additional babysitters.*"

*Mcc*

Mothers in the organization where the mothers themselves care for the children showed less concern about the way their children are cared for. They said they are happy that their children are cared for by themselves and by mothers who have their kids in the center.

"*I feel lucky that I get to spend the whole day with my baby. I care for all of the children as my own. I think this arrangement works for the children and us the mothers pretty well.*"

*Mcc*

**Subtheme five- view on working environment.** Mothers with neither onsite CCC nor six months' maternity leave mentioned that the current maternity leave is not in line with the global EBF recommendation since it is short and unfair.

"*We are advised to exclusively breastfeed up to 6 months for our children to remain healthy. We cannot do that because we return to the office just after 3months.*"

*Mn*

The other issue raised is the concern mothers have related to the implementation of the three months maternity leave which consists of one month before and two months after delivery. In some cases, mothers are forced to take the leave as aforementioned and it's not even allowed for them to take the three months after delivery unless the employer agrees to postpone.

"*The thing is that there needs to be the willingness of the organization to take all of my leave after birth. For example, if I am told to take one month before giving birth as a rule, then I can't do anything about it. Some organizations are flexible and allow mothers to take their three months' maternity leave after giving birth which would motivate us.*"

*Mn*

Mothers pointed out the positive and negative sides of the two options/supporting conditions provided and the responses fell into two categories.

Those supporting the onsite CCC said that it decreases the fear of having no one to take care of a child, brings a sense of security, and ensures better breastfeeding and mental satisfaction.

"*If there is an onsite CCC you will not be stressed and you can breastfeed your baby frequently because the child will be with you. Both your job and child will be benefited and we (mothers) will be healthy and refreshed. I say it is an excellent option.*"

*Mn*

Mothers who have access to onsite CCC talked about how the presence of the center helped them breastfeed and take care of their child better. They said that the center helps them to breastfeed better, caused them no stress, and makes them happy and stable. They also discussed their children being confident, active, and healthier than before and the fact that they would not continue breastfeeding if it wasn't for the center.

"*My child used to be shy but now she is not afraid of people and she wants to play with everyone, she has become confident. There is nothing that could hurt them here it is comfortable and not dangerous at all.*"

*Mcc*

The second category response to CCC was that mothers raised concerns about the type of mother's working environment especially health-related areas, childcare, and the struggle mothers might face when bringing babies to the center and probable loss of focus on their job.

"*In my opinion, CCC brings a lot of hustle to mothers. You have to bring your baby in the morning and return in the evening which is a lot of work. We come from far so it is exhausting both for the mother and child. I also think it will be hard to give 100 percent with your baby being close by. So I'm more inclined to having six months' maternity leave.*"

*Mn*

Regarding the option of six months' maternity leave, some mothers were in favor while some were not for different reasons.

Mothers supported the six months maternity leave mainly because they believe that it facilitates better breastfeeding practice, brings more stability to the mother, and facilitates better childcare.

"*I gave my child formula milk at 3 months because I had only three months' leave and that is bad for my baby. So having six months leave would mean complying with the recommendation of six months of exclusive breastfeeding.*"

*Mn*

Mothers who work in an organization that offers a six months maternity leave expressed their positive experience. Some of the favorable experiences mentioned are an opportunity to better breastfeeding habits, feeling refreshed, and happy. They also expressed that their children are happier and attentive.

"*To say that I could continue to breastfeed without this leave is a joke. If I didn't have this leave I would have started giving my baby other foods early and she would not have been this happy or attentive.*"

*Mn*

Related to longer maternity leave concerns raised by mothers is the probability of decrement of female employment and fear of what could happen after the leave ends.

"*Well, organizations might not benefit much from this (six months' maternity leave) so what will happen is that women's employment will decrease in time. Giving six months would be challenging for some businesses and consequently, the organizations may lose interest to hire women.*"

*Mn*

**Theme two- addressing barriers.** This theme focuses on the coping mechanism that mothers used to breastfeed better and their idea of where the final solution should come from.

**Subtheme one- annual leave as a key.** Almost all mothers with no supporting condition described the use of annual leave to stay longer and take care of their children. Some mothers also described the use of spare time at work to breastfeed and stressed the fact that three months is not enough.

"*I knew I would not be able to work effectively or breastfeed my child properly if I came in to work at three months. So I had to use my entire annual leave to stay longer. You know people don't have to be a woman to feel our pain, everybody should understand that the three months is not enough by any measure.*"

*Mn*

**Subtheme two- lasting resolution.** Finally, mothers emphasized the fact that change should come from the government and that they believe individual organizations would not bring that much difference alone.

"*The change should come nationally from the government or else organizations and bureaus won't allow it and improvement in the EBF rate would not be practical.*"

*Mn*

## Discussion

Actions to improve exclusive breastfeeding have been taken from time to time but studies show that women find it difficult to stick on to expert recommendations of optimal breastfeeding.

The findings in this study showed the context of the working environment and employed mothers' breastfeeding practice that helps to understand the situation in Ethiopia.

Women's attitude and intention towards breastfeeding are affected by different factors at the individual, group, and society level where women employment summed up with the condition of working environment like the length of maternity leave are some of the repeatedly mentioned factors affecting breastfeeding [1,33–35].

This study showed that though most mothers face difficulty while continuing breastfeeding emotional and physical challenges are more intensely experienced by mothers who have none of the supporting conditions. This is in line with a study by Sousan et al which reported that mothers experience emotional stress like guilt and feeling sinful for not breastfeeding well [36]. Difficulties related to lack of facilities like a refrigerator are expressed by mothers who have none of the supporting conditions which is concordant with the *Malaysian* study that pointed out the importance of facilities for maintaining breastfeeding among employed mothers [37].

Mothers in this study expressed that effectiveness and stability at work become possible when they have support from their boss and colleagues which is in agreement with the West Sumatera report that showed working mothers need their superior's assistance to sustain work and exclusively breastfeed [38]. Children are healthier and mothers stable after getting supporting conditions. This is also explained in a study by Brown et al where employers recognized that mothers missed less work when their children are healthy and that breastfed babies tend to be healthier than formula-fed babies while also stressing it as a positive input for organizations [39].

Mothers who have none of the supporting conditions at their workplace believe that CCC and six months of maternity leave present continued breastfeeding, happiness, and stability as major advantages. This is parallel with that of another study where mothers said that it is impossible to breastfeed during shift work without daycare service indicating the need for a child care center [36]. There was also concern about how a mother would be able to bring their children to CCC where there is a difficulty of transportation given their environmental conditions. This should be an area of focus since most mothers in urban settings come from far and live in a highly crowded area.

Mothers raised their concern with a probable decrement of female employee recruitment because of the impact of six months of maternity leave on the operation of organizations; This is supported by a report of a backward slide of mothers' careers, reduced post-birth wages, decreased job opportunity and job uncertainties of women after return to work [40].

Early introduction of formula milk to children is one of the effects of getting back to work after three months of maternity leave which is similar to a study done on nurse mothers where all participants started weaning diet at four months because they had to leave their babies while returning to work [41].

In addition to supporting conditions at the workplace, the type of working area was one point of concern for mothers. Consequently, health facilities were one of the working areas that are mentioned by mothers as being risky to have children stay-in which is also shown by another study where mothers working in a hospital felt that bringing babies in the unit is too risky because wards are not safe [41].

Likelihood of exclusively breastfeeding, having a healthier child, and decreased tendency of early formula milk introduction are related to having supporting conditions at the workplace.

Mothers with six months of maternity leave and onsite CCC tend to expectedly make the supporting condition they currently have their choice of preference.

The three months' leave is expressed as not being sufficient and that it forces mothers to seek other ways of solution like using their annual leave, sick leave, or asking for unofficial arrangements from their bosses. This situation has created a gap that allows and makes employers give unofficial and non-uniform supports to breastfeeding mothers which are

explained in a great intensity by mothers with only three months of maternity leave. Besides, the three months of maternity leave is implemented in a non-consistent way in organizations where it is flexible to postpone in some places but not in others. This inconsistency of the maternity leave would lead to partiality or differential treatment as reflected by the mothers.

Mothers in this study have used their annual leave as a solution and stayed home longer than their officially provided leave. Such type of unexpected longer leave might bring more pressure and affect the work process more than a planned one.

Satisfaction at work is more profoundly expressed by mothers with access to CCC who have scheduled arrangements to provide care for their children. This arrangement can also decrease the concern of employers regarding the cost of hiring additional nannies.

In general, mothers in institutions that offer either of the supporting conditions prefer the support they currently have while mothers with neither of the supporting condition expressed the need of getting supporting conditions at work weighing the different pros and cons of the different types of supporting measures. This shows the need to find a suitable solution that works best for all with minimum shortcomings.

## Conclusions

Providing mothers with a friendly environment makes them work with stability, motivation, and satisfaction based on the current study. This, however, requires a suitable supporting condition with a focus on the different kinds of work environments of the mothers and the different risks related to each respective environment. The presented supporting conditions had their advantages and drawbacks suggesting that there is not a single absolute solution.

Concerning continuous breastfeeding and better care for children, onsite CCC is a better solution since the center provides breastfeeding breaks for mothers to ensure continuous breastfeeding in addition to better childcare. Nonetheless, applying this supporting condition needs preparation to prevent disease transmission among children and avoid extra burden on mothers and ensure that it is not too costly.

Additionally, six months maternity leave was also stated as a good option that can improve breastfeeding habits among working mothers and one that brings more stability to mothers' working status. It is also stressed that it affects mothers' and children's health positively. On the contrary, the probability of decrement of female employment and fear of what could happen after the leave ends is raised as a concern that might need prior preparedness.

Further large-scale quantitative researches are essential to substantiate the evidence generated from this study and look into the cost analysis of the different supporting conditions.

## Study limitation

Due to the shortage of institutions with the workplace supporting arrangements stated in this study, we were not able to include more institutions. Thus, only mothers from the urban setting have been interviewed and this might have a negative impact on presenting more diverse ideas.

## Supporting information

**S1 File. Participant information and interview guide.**
(DOCX)

**S2 File. Themes, sub-themes and codes.**
(DOCX)

**S3 File. Interview data.**
(DOCX)

## Acknowledgments

We would like to thank Addis Ababa University for facilitating this research. We would also like to thank all participants for their input and collaboration in this research.

## Author Contributions

**Conceptualization:** Firmaye Bogale Wolde.

**Formal analysis:** Firmaye Bogale Wolde, Jemal Haidar Ali.

**Investigation:** Firmaye Bogale Wolde.

**Methodology:** Firmaye Bogale Wolde, Jemal Haidar Ali, Yalemwork Getnet Mengistu.

**Project administration:** Firmaye Bogale Wolde.

**Resources:** Firmaye Bogale Wolde.

**Supervision:** Jemal Haidar Ali, Yalemwork Getnet Mengistu.

**Validation:** Firmaye Bogale Wolde, Jemal Haidar Ali, Yalemwork Getnet Mengistu.

**Visualization:** Firmaye Bogale Wolde.

**Writing – original draft:** Firmaye Bogale Wolde.

**Writing – review & editing:** Firmaye Bogale Wolde, Jemal Haidar Ali, Yalemwork Getnet Mengistu.

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
