## [Decision Letter · Decision Letter 0]

8 Apr 2021

PONE-D-20-40733

Employed mothers’ breastfeeding: Exploring breastfeeding experience of employed mothers in different work environments

PLOS ONE

Dear Dr. Wolde,

Thank you for submitting your manuscript to PLOS ONE. After careful consideration, we feel that it has merit but does not fully meet PLOS ONE’s publication criteria as it currently stands. Therefore, we invite you to submit a revised version of the manuscript that addresses the points raised during the review process.

We look forward to receiving your revised manuscript.

Kind regards,

Elena Ambrosino

Academic Editor

PLOS ONE

Journal Requirements:

2. When reporting the results of qualitative research, we suggest consulting the COREQ guidelines: http://intqhc.oxfordjournals.org/content/19/6/349. In this case, please consider including more information on the interviewer’s training and characteristics; and please provide the interview guide used.

4. We note you have included a table to which you do not refer in the text of your manuscript. Please ensure that you refer to Table 1 in your text; if accepted, production will need this reference to link the reader to the Table.

Reviewers' comments:

Reviewer's Responses to Questions

**Comments to the Author**

1. Is the manuscript technically sound, and do the data support the conclusions?

Reviewer #1: Yes

Reviewer #2: Partly

Reviewer #3: Yes

2. Has the statistical analysis been performed appropriately and rigorously? 

Reviewer #1: N/A

Reviewer #2: I Don't Know

Reviewer #3: N/A

3. Have the authors made all data underlying the findings in their manuscript fully available?

Reviewer #1: Yes

Reviewer #2: Yes

Reviewer #3: No

4. Is the manuscript presented in an intelligible fashion and written in standard English?

Reviewer #1: Yes

Reviewer #2: Yes

Reviewer #3: Yes

5. Review Comments to the Author

Reviewer #1: Overall, the manuscript presented an important aspect of child health and wellbeing as well as human rights issue. Through the qualitative descriptive analysis, it has presented factors related to motivation and berries of breastfeeding in working environment nicely. However, there has some issues should be taken care of this manuscript so that readers can understand the information clearly.

No major issues. However the following minor issues should be addressed.

a) Line: 75-78 / I think your research questions and specific objective of this manuscript are similar. If not, please mention about the specific objectives.

b) Line: 97-98/ it would be good to make a separate paragraph about interview guide and questions.

c) Line: 101-103 / about the demographic information collection, you could write in the paragraph of participants.

d) Line: 117-122/ This paragraph can be taken as first paragraph for consistency of the information.

e) Line: 128/ did you transcript it in local dialect or translated into other language? Which language ? Please make it clear.

f) Line: 142/ what is Amharic ? if it is local language please make clear the meaning in the bracket.

g) Did you use any qualitative data analysis software for data coding? If you do please mention name and version of the software.

h) Line: 154-155/ 64% mother’s had 1 child; did 64% mother also belong to mean age 30 years? If different please write different line.

What about the status of other percentage participants’ status? You should mention both.

i) Line: 166-169/ Can you give the full abbreviations of the word mn, mcc and msm?

j) Line: 194/ Did women perceived that there previous children were not in good health because of lack of breast feeding. How did they know about it?

You have collected data from employees of both government and private organizations. What about the government policy about maternity leave in Addis Ababa? You could mention it in the discussion section.

Reviewer #2: It is an important topic especially as more women enter the workforce. Please see suggested changes/questions

Line 18/19- please mention a source of the literature

Line 40- please start the sentence with breast feeding is a life saving intervention. It is one of the most cost-effective interventions to reduce infant mortality and that point should be made upfront

Line 43 Please mention WHO recommendations around exclusive breastfeeding and maybe look for a more recent source. Authors are quoting from 2000

Line 47 please look for data that demonstrates the link of breastfeeding with Infant mortality in addition to child mortality

Line 54-59. Suggest moving this paragraph at the beginning – around line 43

Line 63 and 64- is there a source for these recommendations? What about breastfeeding rooms at workplace where women can pump and store breastmilk? Why is that not referenced?

Line 80- please explain why this design and approach was selected and why is it most suitable to answer the questions being studied

In the study design, please indicate the rationale for the sample size. Is interviewing 17 mothers sufficient? What is the guidance?

In the background, please mention if Ethiopia has any nation-wide policies that guide maternity leave or child care.

In the conclusion- I don’t see any reference to the option of 6 month maternity leave. It seems that women quoted that as a good option. The only one highlighted is the child care center.

Please include a section on study limitations. The sample size seems extremely small to be able to draw your conclusions

Reviewer #3: In the manuscript, Employed mothers’ breastfeeding: Exploring breastfeeding experience of employed mothers in different work environments, the authors conduct a series of in-depth interviews with employed women in Ethiopia who have varying levels of postpartum breastfeeding support – three months of maternity leave, six months of maternity leave, and onsite child care. Emergent themes included the barriers and facilitators to breastfeeding while returning to work, the impact of the workplace environment on the breastfeeding journey, and ways in which women cope with challenges associated with breastfeeding while employed. The study results speak to the lived experiences of women in Ethiopia who work in various employment settings while continuing to breastfeed. The manuscript would benefit from additional edits, please see below for specific suggestions.

Major suggestions include:

(1) The impact of employment, or return to employment, on breastfeeding is quite will researched throughout various contexts. In the Introduction, the authors state that few studies “focus on exploring the opinions and experiences of employed mothers on breastfeeding…” It seems that there are, however, quite a few qualitative and quantitative studies exploring women’s experiences with employment and breastfeeding. After an initial search through the literature, it seems that research being published on the topic specific to Ethiopia is quantitative. I might suggest framing this study within the context of what is currently known about Ethiopia specifically, rather than including a mix of global and country-specific data in the Introduction. For example, the authors could consider framing this study as a way to better understand the quantitative findings and trends that exist in Ethiopia through a qualitative exploration of women’s lived experiences. To achieve this, I would suggest tailoring the introduction to be more specific to the study context; after summarizing the importance of breastfeeding globally, the authors could describe what is known broadly – and within Ethiopia specifically – about the barriers and facilitators to breastfeeding among employed women. The authors could then describe existing gaps in the literature specifically within the context of what is known on the topic in Ethiopia, rather than globally. I would also suggest this change is reflected in the title of the manuscript, making it clear that the findings are specific to the study context rather than more globally generalizable.

(2) The authors describe the inclusion and selection criteria for study participants. However, it would be useful to include additional detail on the sampling procedure, such as how institutions were selection and what communication with participants entailed for recruitment and follow-up.

(3) The authors include an overview of the various phases of data analysis; at the end of the Data Analysis section (Lines 145-148), the authors explain that all three authors were included in each phase of analysis. I would suggest moving this statement up in the section and providing additional detail to make it clear who was involved in each phase of analysis and what their roles were.

(4) The authors present very rich, insightful data as a result of the interviews; the quotes included are very powerful. When reading the Results section, I have a difficult time identifying the differences between the first two themes – motivators and barriers, and work environment and breastfeeding. There also seems to be some overlap in content between these two main themes; for example, within both themes there’s a discussion around women’s experiences differing between their children. In the third theme, coping mechanisms, the data from the quotes indicate that the authors collected many suggestions from women on how to improve their experiences, as well as suggestions around larger, systemic improvements. It seems like the two main categories of themes might be related to (1) barriers and facilitators to breastfeeding and employment, and (2) suggestions for addressing barriers. It might be helpful to consider how to re-structure the Results section in a way that tells a clearer story of women’s experiences for the reader.

(5) Similar to my suggestions related to the Introduction, it would be useful to consider re-framing the Discussion section to first explain what these study findings add to the generalizable knowledge within Ethiopia, and then include a broader assessment of how these findings relate to other contexts. For example, a qualitative study was recently published (Gebrekidan, et al. 2020) describing managers’ perspectives on factors that impact exclusive breastfeeding in the work place in Ethiopia; consider situating these results within what is known specifically within Ethiopia.

Minor suggestions include:

(1) Under Data Collection, the authors explain that the data collection tool used for interviews was adapted from a national breastfeeding study in Pakistan. Please include a citation for this study, as well as related citations for its application in other countries.

(2) The “Operational Definition” section seems like it may be out of place. Can this be included elsewhere in the narrative, without having a separate section to define what is meant by a supportive condition at work?

(3) Include a reference to Table 1 within the text of the manuscript.

(4) Please be sure that the manuscript formatting aligns with the journal requirements.

6. PLOS authors have the option to publish the peer review history of their article (what does this mean?). If published, this will include your full peer review and any attached files.

Reviewer #1: No

Reviewer #2: No

Reviewer #3: **Yes: **Stephanie Bogdewic

---

## [Author Response · Author response to Decision Letter 0]

30 Jul 2021

Reviewer 1: 

Overall, the manuscript presented an important aspect of child health and wellbeing as well as human rights issue. Through the qualitative descriptive analysis, it has presented factors related to motivation and berries of breastfeeding in working environment nicely. However, there has some issues should be taken care of this manuscript so that readers can understand the information clearly. No major issues. However the following minor issues should be addressed.

1. Line: 75-78 / I think your research questions and specific objective of this manuscript are similar. If not, please mention about the specific objectives

Authors’ response: Thank you for your comment. Even though the research question and the objectives seem similar there are more details in the specific objectives. Thus, we have included the specific objectives in the first section of the materials and methods and stated the specific objectives in (line 92-96)

2. Line: 97-98/ it would be good to make a separate paragraph about interview guide and questions

Authors’ response: as per the suggestion we have made the paragraphs separate in the data collection section (second and third paragraph)

3. Line: 101-103 / about the demographic information collection, you could write in the paragraph of participants

Authors’ response: while we appreciate the reviewer’s suggestion, we feel that the collected data is better placed in the data collection section with the explanation of the question types. But we have rearranged the place (line 135-137)

4. Line: 117-122/ This paragraph can be taken as first paragraph for consistency of the information

Authors’ response: we agree with the comment and we have moved it to the first paragraph (line 124-129)

5. Line: 128/ did you transcript it in local dialect or translated into other language? Which language? Please make it clear

Authors’ response: Thank you for pointing this out. We have clarified that Amharic is a local dialect (line 166)

6. Line: 142/ what is Amharic? If it is local language please make clear the meaning in the bracket

Authors’ response: in relation with the above comments we have clarified that Amharic is a local dialect (line 166)

7. Did you use any qualitative data analysis software for data coding? If you do please mention name and version of the software

Authors’ response: we did the analysis manually so we did not use software.

8. Line: 154-155/ 64% mother’s had 1 child; did 64% mother also belong to mean age 30 years? If different please write different line. What about the status of other percentage participants’ status? You should mention both

Authors’ response: According to the suggestion by the reviewer we have seen that the statement could be confusing and we have edited it to include additional information and be clearer. Result section (line 178-184)

9. Line: 166-169/ Can you give the full abbreviations of the word mn, mcc and msm?

Authors’ response: we have added the suggested full abbreviations for Mn, Mcc and Msm (line 188-195)

10. Line: 194/ Did women perceived that there previous children were not in good health because of lack of breast feeding. How did they know about it?

Authors’ response: Thank you for the question. As indicated in the quoted statement of a mother on line 225-227 other mothers also stated that they saw the difference in the frequency of illness and status of health of their breastfed and non-breastfed child which made them almost certain that the breastfeeding brought change on the health and immunity of their children

11. You have collected data from employees of both government and private organizations. What about the government policy about maternity leave in Addis Ababa? You could mention it in the discussion section

Authors’ response: Thank you for the comment. We agree that it is an important aspect to show. Thus, we have included it in the background section (line 75-77)

 

Reviewer 2:

It is an important topic especially as more women enter the workforce. Please see suggested changes/questions

1. Line 18/19- please mention a source of the literature

Authors’ response: Thank you for the comment. While we respect the suggestion, we feel it is better to cite the source in the background section (line 62) where we brought this statement from.

2. Line 40- please start the sentence with breast feeding is a life-saving intervention. It is one of the most cost-effective interventions to reduce infant mortality and that point should be made upfront

Authors’ response: Thank you for the insightful comment and we have edited the start of the introduction as recommended (line 39-42)

3. Line 43 Please mention WHO recommendations around exclusive breastfeeding and maybe look for a more recent source. Authors are quoting from 2000

Authors’ response: thank you for the comment. We have mentioned the WHO recommendation for exclusive breastfeeding on line 43-46 and we have also corrected the WHO report to a more recent source by the year 2021 (line 47-49) 

4. Line 47 please look for data that demonstrates the link of breastfeeding with Infant mortality in addition to child mortality

Authors’ response: we appreciated the comment and we have included studies that show the relation of breastfeeding with neonatal, infant and under-five child mortality for a wholesome view (line50-52)

5. Line 54-59. Suggest moving this paragraph at the beginning – around line 43

Authors’ response: We accept the suggestion and moved the statement to first paragraph in the introduction and edited the statements 

6. Line 63 and 64- is there a source for these recommendations? What about breastfeeding rooms at workplace where women can pump and store breastmilk? Why is that not referenced?

Authors’ response: Thank you for an insightful comment. We have included additional settings for mother friendly worksites in background section in (line 57-62) and (line 71-73)

7. Line 80- please explain why this design and approach was selected and why is it most suitable to answer the questions being studied

Authors’ response: Thank you for pointing the missing information. We have explained the reason for the selection of the design (line 102-106)

8. In the study design, please indicate the rationale for the sample size. Is interviewing 17 mothers sufficient? What is the guidance?

Authors’ response: We appreciate the suggestion. The rationale for the number of participants was information saturation. Accordingly, we have included the explanation for the number of study participants being based on information saturation in (line 127-129). 

9. In the background, please mention if Ethiopia has any nation-wide policies that guide maternity leave or child care

Authors’ response: Thank you for the important comment. Accordingly, we have included it in the background section (line 75-77)

10. In the conclusion- I don’t see any reference to the option of 6 month maternity leave. It seems that women quoted that as a good option. The only one highlighted is the child care center

Authors’ response: Thank you for pointing this out. We have now highlighted the six months maternity leave in conclusion (line 466-470)

11. Please include a section on study limitations. The sample size seems extremely small to be able to draw your conclusions

Authors’ response: Thank you for the comment. We have tried as much as possible to attain information saturation during data collection and we have based our conclusion on the extensive information from the interviewed participants. However, it is true that we were not able to include mothers from diverse organizations which had supporting arrangements for mothers which has in turn limited the diversity of women and ideas included and thus we have agreed to include study limitation presented in (line 474-477)

 

Reviewer 3:

In the manuscript, Employed mothers’ breastfeeding: Exploring breastfeeding experience of employed mothers in different work environments, the authors conduct a series of in-depth interviews with employed women in Ethiopia who have varying levels of postpartum breastfeeding support – three months of maternity leave, six months of maternity leave, and onsite child care. Emergent themes included the barriers and facilitators to breastfeeding while returning to work, the impact of the workplace environment on the breastfeeding journey, and ways in which women cope with challenges associated with breastfeeding while employed. The study results speak to the lived experiences of women in Ethiopia who work in various employment settings while continuing to breastfeed. The manuscript would benefit from additional edits, please see below for specific suggestions.

Major suggestions include 

1. The impact of employment, or return to employment, on breastfeeding is quite will researched throughout various contexts. In the Introduction, the authors state that few studies “focus on exploring the opinions and experiences of employed mothers on breastfeeding…” It seems that there are, however, quite a few qualitative and quantitative studies exploring women’s experiences with employment and breastfeeding. After an initial search through the literature, it seems that research being published on the topic specific to Ethiopia is quantitative. I might suggest framing this study within the context of what is currently known about Ethiopia specifically, rather than including a mix of global and country-specific data in the Introduction. For example, the authors could consider framing this study as a way to better understand the quantitative findings and trends that exist in Ethiopia through a qualitative exploration of women’s lived experiences. To achieve this, I would suggest tailoring the introduction to be more specific to the study context; after summarizing the importance of breastfeeding globally, the authors could describe what is known broadly – and within Ethiopia specifically – about the barriers and facilitators to breastfeeding among employed women. The authors could then describe existing gaps in the literature specifically within the context of what is known on the topic in Ethiopia, rather than globally. I would also suggest this change is reflected in the title of the manuscript, making it clear that the findings are specific to the study context rather than more globally generalizable.

Authors’ response: We sincerely appreciate the depth of review and comment. As suggested in the comment we have tried to focus the introduction to the Ethiopian context, what is missing and the current proclamation on maternity leave. Additionally, we have edited the title to indicate its focus by including the term “Ethiopia” in the title.

2. The authors describe the inclusion and selection criteria for study participants. However, it would be useful to include additional detail on the sampling procedure, such as how institutions were selection and what communication with participants entailed for recruitment and follow-up

Authors’ response: Thank you for the comment. We have accepted the suggestion and elaborated more on the selection of institutions and participants for the study in Participants section (line 113-122)

3. The authors include an overview of the various phases of data analysis; at the end of the Data Analysis section (Lines 145-148), the authors explain that all three authors were included in each phase of analysis. I would suggest moving this statement up in the section and providing additional detail to make it clear who was involved in each phase of analysis and what their roles were

Authors’ response: we appreciate the comment and even though all authors were involved in the study we have elaborated their specific participation in (line 158-164) 

4. The authors present very rich, insightful data as a result of the interviews; the quotes included are very powerful. When reading the Results section, I have a difficult time identifying the differences between the first two themes – motivators and barriers, and work environment and breastfeeding. There also seems to be some overlap in content between these two main themes; for example, within both themes there’s a discussion around women’s experiences differing between their children. In the third theme, coping mechanisms, the data from the quotes indicate that the authors collected many suggestions from women on how to improve their experiences, as well as suggestions around larger, systemic improvements. It seems like the two main categories of themes might be related to (1) barriers and facilitators to breastfeeding and employment, and (2) suggestions for addressing barriers. It might be helpful to consider how to re-structure the Results section in a way that tells a clearer story of women’s experiences for the reader.

Authors’ response: Thank you again for the insightful comment. After reading and rereading the themes and subthemes we also agree that that they can be brought together. Thus, we have incorporated the findings in to two themes where the themes are in (line 216 and line 372) 

5. Similar to my suggestions related to the Introduction, it would be useful to consider re-framing the Discussion section to first explain what these study findings add to the generalizable knowledge within Ethiopia, and then include a broader assessment of how these findings relate to other contexts. For example, a qualitative study was recently published (Gebrekidan, et al. 2020) describing managers’ perspectives on factors that impact exclusive breastfeeding in the work place in Ethiopia; consider situating these results within what is known specifically within Ethiopia.

Authors’ response: In line with the introduction we have also made some changes in the discussion where we have tried to focus on Ethiopian context and relate it with the global setting (changes in the discussion are highlighted)

Minor suggestions include: 

1. Under Data Collection, the authors explain that the data collection tool used for interviews was adapted from a national breastfeeding study in Pakistan. Please include a citation for this study, as well as related citations for its application in other countries.

Authors’ response: Thank you for pointing that out. We have now included the reference (line 132)

2. The “Operational Definition” section seems like it may be out of place. Can this be included elsewhere in the narrative, without having a separate section to define what is meant by a supportive condition at work?

Authors’ response: we agree with the suggestion and we have included the definition in participants section (line 114-115)

3. Include a reference to Table 1 within the text of the manuscript.

Authors’ response: Thank you, we have cited the table in the manuscript as suggested (line 188)

4. Please be sure that the manuscript formatting aligns with the journal requirements.

Authors’ response: Thank you for stressing this out. We have tried to carefully align the manuscript with journal requirements.

---

## [Decision Letter · Decision Letter 1]

15 Sep 2021

PONE-D-20-40733R1Employed mothers’ breastfeeding: Exploring breastfeeding experience of employed mothers in different work environments in EthiopiaPLOS ONE

Dear Dr. Wolde,

Thank you for submitting your manuscript to PLOS ONE. After careful consideration, we feel that it has merit but does not fully meet PLOS ONE’s publication criteria as it currently stands. Therefore, we invite you to submit a revised version of the manuscript that addresses the points raised during the review process.

We look forward to receiving your revised manuscript.

Kind regards,

Elena Ambrosino

Academic Editor

PLOS ONE

Journal Requirements:

Reviewers' comments:

Reviewer's Responses to Questions

**Comments to the Author**

1. If the authors have adequately addressed your comments raised in a previous round of review and you feel that this manuscript is now acceptable for publication, you may indicate that here to bypass the “Comments to the Author” section, enter your conflict of interest statement in the “Confidential to Editor” section, and submit your "Accept" recommendation.

Reviewer #2: All comments have been addressed

Reviewer #3: (No Response)

2. Is the manuscript technically sound, and do the data support the conclusions?

Reviewer #2: Yes

Reviewer #3: Yes

3. Has the statistical analysis been performed appropriately and rigorously? 

Reviewer #2: Yes

Reviewer #3: N/A

4. Have the authors made all data underlying the findings in their manuscript fully available?

Reviewer #2: Yes

Reviewer #3: Yes

5. Is the manuscript presented in an intelligible fashion and written in standard English?

Reviewer #2: Yes

Reviewer #3: Yes

6. Review Comments to the Author

Reviewer #2: The authors have done a great job of responding to the comments. Please copyedit the document.

Reviewer #3: Thank you for the opportunity to review the updated manuscript titled, "Employed mothers’ breastfeeding: Exploring breastfeeding experience of employed mothers in different work environments in Ethiopia." The authors successfully addressed the majority of reviewer comments. However, the text added to the Introduction to describe the quantitative research done in Ethiopia on the topic is missing citations. Please add citations to support the claims made on lines 79-82.

7. PLOS authors have the option to publish the peer review history of their article (what does this mean?). If published, this will include your full peer review and any attached files.

Reviewer #2: No

Reviewer #3: No

---

## [Author Response · Author response to Decision Letter 1]

19 Oct 2021

Reviewer #2: 

1- The authors have done a great job of responding to the comments. Please copyedit the document.

Authors response:- Thank you for your time and comments. We have now copyedited the document. 

Reviewer #3: 

1- Thank you for the opportunity to review the updated manuscript titled, "Employed mothers’ breastfeeding: Exploring breastfeeding experience of employed mothers in different work environments in Ethiopia." The authors successfully addressed the majority of reviewer comments. However, the text added to the Introduction to describe the quantitative research done in Ethiopia on the topic is missing citations. Please add citations to support the claims made on lines 79-82.

Authors response:- Thank you for your comments and your time. We have now inserted relevant references to the statement on lines 79-82

---

## [Editor Report · Decision Letter 2]

28 Oct 2021

Employed mothers’ breastfeeding: Exploring breastfeeding experience of employed mothers in different work environments in Ethiopia

PONE-D-20-40733R2

Dear Dr. Wolde,

We’re pleased to inform you that your manuscript has been judged scientifically suitable for publication and will be formally accepted for publication once it meets all outstanding technical requirements.

Kind regards,

Elena Ambrosino

Academic Editor

PLOS ONE
---

## [Editor Report · Acceptance letter]

3 Nov 2021

PONE-D-20-40733R2 

Employed mothers’ breastfeeding: Exploring breastfeeding experience of employed mothers in different work environments in Ethiopia 

Dear Dr. Wolde:

I'm pleased to inform you that your manuscript has been deemed suitable for publication in PLOS ONE. Congratulations! Your manuscript is now with our production department. 

Kind regards, 

on behalf of

Dr. Elena Ambrosino 

Academic Editor

PLOS ONE